# SegRet: An Efficient Design for Semantic Segmentation with Retentive Network

## Abstract

With the rapid evolution of autonomous driving technology and intelligent transportation systems, semantic segmentation has become increasingly critical. Precise interpretation and analysis of real-world environments are indispensable for these advanced applications. However, traditional semantic segmentation approaches frequently face challenges in balancing model performance with computational efficiency, especially regarding the volume of model parameters. To address these constraints, we propose SegRet, a novel model employing the Retentive Network (RetNet) architecture coupled with a lightweight residual decoder that integrates zero-initialization. SegRet offers three distinctive advantages: (1) Lightweight Residual Decoder: by embedding a zero-initialization layer within the residual network structure, the decoder remains computationally streamlined without sacrificing essential information propagation; (2) Robust Feature Extraction: adopting RetNet as its backbone enables SegRet to effectively capture hierarchical image features, thereby enriching the representation quality of extracted features; (3) Parameter Efficiency: SegRet attains state-of-the-art (SOTA) segmentation performance while markedly decreasing the number of parameters, ensuring high accuracy without imposing additional computational burdens. Comprehensive empirical evaluations on prominent benchmarks, such as ADE20K, Citycapes, and COCO-Stuff, highlight the effectiveness and superiority of our method.

## 1 Introduction

Semantic segmentation, which assigns semantic labels to every pixel in an image, constitutes a fundamental problem in computer vision (Lateef & Ruichek, 2019). It serves as the foundation for critical applications such as autonomous driving, urban scene analysis, and intelligent transportation systems, where accurate perception of dynamic environments is indispensable for safety and efficiency (Hao et al., 2020; Zhao et al.). Despite remarkable advances, achieving a favorable balance between segmentation accuracy and computational efficiency remains an open challenge.

Classical deep learning approaches, including Fully Convolutional Networks (FCN) (Long et al., 2014), Mask R-CNN (He et al., 2017), and Pyramid Scene Parsing Network (PSP-Net) (Zhao et al., 2017), have significantly advanced the state of the art by introducing powerful encoder–decoder paradigms. More recently, Transformer-based architectures, such as the Vi-

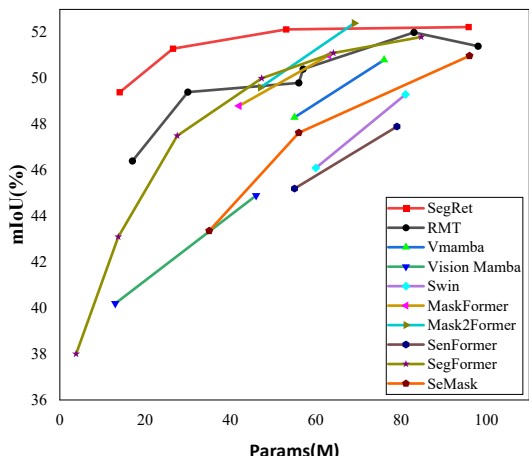

Figure 1: Performance comparison on ADE20K. Multi-scale inference is used for all results.

sion Transformer (ViT) (Dosovitskiy et al., 2021), Swin Transformer (Liu et al., 2021), and Detection Transformer (DETR) (Carion et al., 2020), have demonstrated outstanding capabilities in modeling long-range dependencies. Nevertheless, their reliance on quadratic-complexity self-attention, coupled

with large parameter counts, severely limits their practicality in real-time and resource-constrained scenarios. To mitigate these drawbacks, a line of efficient architectures has been proposed, including ReViT (Diko et al., 2024), ExMobileViT (Yang et al., 2023), and EfficientFormer (Li et al., 2022), which reduce computational redundancy by introducing localized attention mechanisms or lightweight convolutions. However, these methods often entail a compromise between efficiency and accuracy.

Concurrently, state-space models have emerged as competitive alternatives to Transformers. Notable examples include Mamba (Gu & Dao, 2023), RWKV (Li et al., 2024b), and RetNet (Sun et al., 2023), which achieve linear scalability while retaining the ability to capture long-range dependencies. Their successful adaptation to the visual domain has led to a new family of architectures, such as VMamba (Liu et al., 2024b), VM-UNet (Ruan & Xiang, 2024), and RSMamba (Chen et al., 2024a), which demonstrate promising results across a variety of vision tasks. Nevertheless, within the semantic segmentation domain, most encoder–decoder frameworks still rely on parameter-heavy decoders. Architectures such as UperNet (Xiao et al., 2018) and MaskFormer (Cheng et al., 2021) exemplify this issue, introducing substantial computational overhead that hinders deployment in real-time scenarios.

In this paper, we present **SegRet**, an efficient semantic segmentation framework that integrates Vision RetNet (Sun et al., 2023) as a hierarchical encoder with a lightweight zero-initialized residual decoder. Specifically, Vision RetNet, pre-trained on ImageNet1K (Deng et al., 2009), provides robust multi-scale feature extraction without the computational burden of global self-attention. Complementarily, our decoder introduces a novel zero-initialized residual design that enables effective hierarchical feature fusion with minimal parameter overhead.

Our contributions can be summarized as follows:

- We leverage Vision RetNet as a hierarchical encoder to capture rich multi-scale features efficiently, mitigating the computational limitations of Transformer-based backbones.

- We propose a lightweight residual decoder with zero-initialization that achieves effective feature fusion while significantly reducing the parameter count compared to conventional decoders.

- We conduct extensive experiments on three benchmark datasets—ADE20K (Zhou et al., 2017), Cityscapes (Cordts et al., 2016), and COCO-Stuff (Caesar et al., 2018)—demonstrating that SegRet achieves favorable performance with superior parameter efficiency, thereby offering a practical solution for real-world deployment.

## 2 RELATED WORK

### 2.1 SEMANTIC SEGMENTATION.

Traditional semantic segmentation approaches primarily depend on manually engineered features and classifiers for pixel-level classification. These methods typically employ low-level features such as color, texture, and shape, coupled with algorithms like graph cuts and random forests, to perform segmentation tasks (Zheng et al., 2012; Arbelaez et al., 2010; Zhang et al., 2016). However, due to the inherent complexity and variability of real-world scenarios, traditional approaches often fail to adequately manage challenges such as occlusion and varying illumination, resulting in inaccurate segmentation outcomes.

The advent of deep learning has notably enhanced the performance of semantic segmentation. Deep learning models facilitate mapping from pixel-level features to semantic labels through upsampling and skip connections, markedly improving both accuracy and computational efficiency (Paszke et al., 2016; Yu et al., 2017; He et al., 2019a;b). Contemporary models predominantly utilize Transformer-based architectures, incorporating self-attention mechanisms to effectively capture global pixel dependencies, thereby improving the understanding and representation of semantic information within images (Zheng et al., 2021; Xie et al., 2021; Cheng et al., 2022; Zhang et al., 2022). Recent advancements integrate zero-shot learning and prompt learning strategies into semantic segmentation, enabling the identification of new object classes without relying on annotated datasets and offering additional contextual guidance. These innovative approaches exhibit significant

advantages in addressing challenges associated with data scarcity and further enhancing segmentation performance (Kirillov et al., 2023; Zhang et al., 2023; 2024b).

## 2.2 STATE SPACE MODEL

Given the limitations of the self-attention mechanism in effectively handling long text sequences, researchers have turned their attention toward alternative architectures to address this challenge. In particular, state-space models such as Mamba (Gu & Dao, 2023) and RetNet (Sun et al., 2023) have attracted considerable interest. Unlike Transformers, these architectures employ state-space modeling mechanisms tailored explicitly for sequence processing. Specifically, Mamba utilizes a selective state-space approach that achieves linear computational complexity, while RetNet employs recurrent state aggregation combined with three computational paradigms to robustly capture long-range dependencies. These advancements offer significant improvements in both efficiency and performance for long-sequence processing tasks compared to Transformer-based models.

Extending the proven efficacy of state-space models into visual domains, researchers have introduced several visual architectures based on these principles. These visual models not only inherit the capacity of state-space methods to manage extended sequences but also integrate distinctive aspects of visual data processing, enabling them to effectively handle complex visual information from images and videos. Prominent among these are Vmamba (Liu et al., 2024b), Vision Mamba (Zhu et al., 2024), and Vision RetNet (Fan et al., 2023). These models have demonstrated notable success across various tasks, including image classification and object detection. Additionally, they offer innovative approaches for addressing intricate visual tasks such as video comprehension in dynamic contexts, medical image analysis involving multi-modal data fusion, and the detection of small infrared targets in low-contrast scenarios (Li et al., 2024a; Liu et al., 2024a; Chen et al., 2024b).

## 3 METHOD

This section presents the SegRet model, initially providing an overview of its foundational architecture. Following this, we delve into a comprehensive discussion of Vision RetNet, highlighting its effectiveness as a hierarchical feature extractor for robust feature representation. We then thoroughly examine the lightweight residual decoder, underscoring its crucial role in maintaining model accuracy while significantly reducing the number of parameters.

As illustrated in Figure 2, the SegRet model adopts an encoder-decoder structure. The encoder employs Vision RetNet to extract hierarchical features at four distinct resolutions. These multi-scale features are subsequently processed by the residual decoder, which comprises a linear mapping block and a zero-initialized residual block. After feature fusion within this decoder, the output is transformed into a semantic segmentation mask.

### 3.1 VISION RETNET BACKBONE

#### 3.1.1 RETNET

Firstly, we revisit the self-attention mechanism of the Transformer. For each input vector $X$, by multiplying matrices $W_Q$, $W_K$ and $W_V$ with $X$, we obtain the $Q$ (Query), $K$ (Key) and $V$ (Value) vectors respectively. Therefore, the self-attention is defined as:

$$\text{Attention}(Q, K, V) = Softmax\left(\frac{QK^T}{\sqrt{d_k}}\right)V, \tag{1}$$

where $d_k$ is the dimension of the $K$ vectors.

To overcome challenges associated with training parallelism, efficient inference, and performance optimization commonly encountered by Transformers in complex task scenarios, the Retentive Network (RetNet) architecture has been introduced. RetNet establishes a novel foundational framework that theoretically elucidates the relationship between recurrence and attention mechanisms. Central to RetNet is the innovative retention mechanism tailored specifically for sequence modeling, facilitating three distinct computational modes: parallel, recurrent, and block-recurrent.

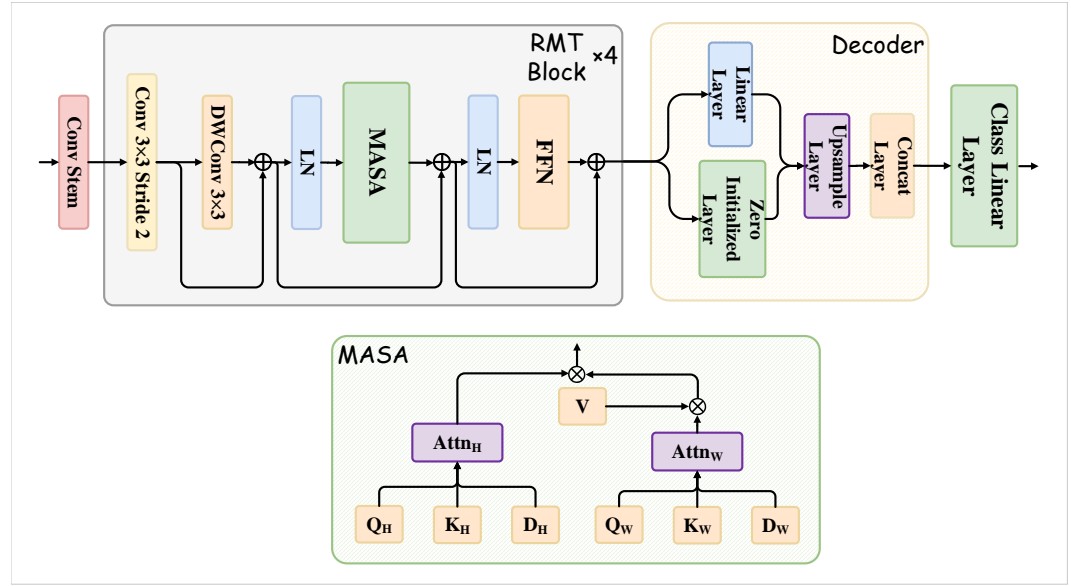

Figure 2: **An overview of the proposed SegRet model.** We use Vision RetNet (RMT) as a hierarchical feature extractor to introduce RetNet into semantic segmentation (Section 3.1). For more efficient feature fusion, a zero-initialization residual decoder is applied to predict semantic segmentation masks (Section 3.2).

The retention mechanism is the core of RetNet, with the following fundamental principles:

$$o_n = \sum_{m=1}^{n} \gamma^{n-m} \left(Q_n e^{in\theta}\right) \left(K_m e^{im\theta}\right)^{\dagger} v_m, \tag{2}$$

where $Q$ and $K$ are input-derived vectors generated through affine transformations. $e^{in\theta}$ and $e^{im\theta}$ serve as rotational factors that encode positional information using complex exponential forms, where $n$ and $m$ represent the positional indices within the sequence, and $\theta$ denotes the learnable parameters employed to model relative phase differences for the purpose of capturing sequential dependencies. $v_m$ stands for the value vector at position $m$, $\gamma$ represents an exponential decay factor, and $\dagger$ signifies the conjugate transpose.

Building upon this foundation, the parallel form is deduced as:

$$Q = (XW_Q) \odot \Theta, K = (XW_K) \odot \bar{\Theta}, V = XW_V, \tag{3}$$

$$\Theta_n = e^{in\theta}, D_{nm} = \begin{cases} \gamma^{n-m}, & n \geq m \\ 0, & n < m \end{cases}, \tag{4}$$

$$Retention(X) = (QK^{\top} \odot D)V, \tag{5}$$

where $\bar{\Theta}$ is the complex conjugate of $\Theta$, $\odot$ denotes the Hadamard product, $D \in \mathbb{R}^{|x| \times |x|}$ represents causal masking and exponential decay, indicating relative distances within a one-dimensional sequence. RetNet's exceptional performance, training parallelism, cost-effective deployment, and efficient inference collectively enable it to successfully mitigate the challenge of excessive computational complexity associated with Transformer.

### 3.1.2 VISION RETNET

Vision RetNet (Fan et al., 2023) is among the pioneering visual state-space models introduced specifically for CV tasks. It enhances the original RetNet architecture by modifying the matrix

$D$, adapting it from a unidirectional form utilized in NLP to a bidirectional form tailored for CV applications. This adaptation effectively manages causal masking and exponential decay, aligning the model's design with the distinct requirements of CV tasks.

Formally, the BiRetention operator is defined as follows:

$$\text{BiRetention}(X) = \left(QK^{\top} \odot D^{Bi}\right)V, \tag{6}$$

$$D^{Bi}_{nm} = \gamma^{|n-m|}, \tag{7}$$

where BiRetention denotes the retention with bidirectional modeling ability.

Specifically, for the two-dimensional spatial attributes of images, the matrix D is modified to its two-dimensional version to better capture spatial relationships, expressed as:

$$D^{2d}_{nm} = \gamma^{|x_n - x_m| + |y_n - y_m|}, \tag{8}$$

where $x$ and $y$ are two-dimensional coordinates in the image.

Moreover, the high resolution typical of visual images generates an extensive number of tokens, significantly increasing computational complexity. To effectively mitigate this issue, Vision RetNet introduces a computational strategy that decomposes processing along both horizontal and vertical image axes. Specifically, it calculates attention mechanisms and distance matrices independently in these two directions, thereby considerably reducing computational overhead. The detailed computational procedure is described as follows:

$$Q_H, K_H = (Q, K)^{B,L,C \to B,W,H,C}, \tag{9}$$

$$Q_W, K_W = (Q, K)^{B,L,C \to B,H,W,C}, \tag{10}$$

$$Attn_H = Softmax(Q_H K_H{}^T) \odot D^H, \tag{11}$$

$$Attn_W = Softmax(Q_W K_W{}^T) \odot D^W, \tag{12}$$

$$D^H_{nm} = \gamma^{|y_n - y_m|}, \quad D^W_{nm} = \gamma^{|x_n - x_m|}, \tag{13}$$

$$ReSA_{dec}(X) = Attn_H(Attn_W V)^T, \tag{14}$$

where $B$ denotes the batch size, $H$ and $W$ indicate the number of non-overlapping patches along the height and width of the input image, respectively, $C$ is the feature dimensionality of each token, and $L = H \times W$ corresponds to the total number of visual tokens obtained by partitioning the image and flattening the patches into one-dimensional embeddings.

As shown in Fig 2, the proposed SegRet model consists of Vision RetNet, which combines the output features of Vision RetNet blocks after each downsampling step into hierarchical feature matrices and feeds them into the decoder for further processing.

### 3.2 LIGHTING RESIDUAL DECODER

Currently, most widely adopted decoders rely heavily on intricate CNN or Transformer architectures, leading to substantial parameter counts and compromised real-time performance. To overcome these limitations, we introduce a lightweight residual decoder. As depicted in Figure 2, the decoder comprises two main components: a linear mapping block and a zero-initialized residual structure. The detailed architecture of our decoder is presented as follows:

The decoder accepts as input a set of features derived from various layers, represented by $\tilde{F} = [f_1, f_2, ..., f_n]$, where each $f_i$ corresponds to features extracted from the $i$-th layer and exhibits different channel dimensions. The primary goal of the decoder is to consolidate these multi-scale features into a coherent output with dimensions $n_{cls} \times H \times W$, where $n_{cls}$ denotes the total number of classes, and $H$ and $W$ specify the output image's height and width, respectively. The decoding procedure is formally described as follows:

Initially, each input feature $f_i$ is subjected to a linear transformation to unify its channel dimension to a common size $C$, yielding the transformed feature $F_i$. This step can be formulated as:

$$F_i = Linear(f_i). \tag{15}$$

To preserve the original information while enhancing feature representation, we apply a residual connection between the original feature $f_i$ and its linearly transformed counterpart $F_i$. This residual connection is realized through a $1 \times 1$ convolutional layer initialized with zeros, ensuring that the output feature map maintains the same channel dimension as $f_i$. Formally, the output of the residual connection layer is defined as:

$$F_i' = f_i + \text{Zero-initialized Conv}(F_i), \tag{16}$$

where zero-initialized Conv denotes a $1 \times 1$ convolution operation with zero initialization. All features $F_i'$ are upsampled to $1/4$ of the image size. The upsampled feature can be represented as $\hat{F}_i = \text{Upsample}(F_i')$. The upsampled features $\hat{F}_i$ are then concatenated to obtain the output image. Assuming the merged feature is $M$, the concatenation operation can be represented as:

$$M = Concat(\hat{F}_1, ..., \hat{F}_n). \tag{17}$$

The dimensions of $M$ are $[4C, H, W]$. Finally, the merged feature $M$ undergoes a convolution mapping to adjust the channel size to the required number of classes $n_{cls}$. This process can be expressed as:

$$Output = Conv(M). \tag{18}$$

## 4 EXPERIMENTS

We conducted comprehensive comparisons with recent SOTA methods on the ADE20K, Cityscapes, and COCO-Stuff datasets, demonstrating the effective integration of RetNet into the semantic segmentation domain through our SegRet model. By evaluating model parameters alongside mean Intersection over Union (mIoU) scores, our experiments substantiate that SegRet is a robust and competitive solution for semantic segmentation tasks.

**Datasets**  ADE20K (Zhou et al., 2017), Cityscapes (Cordts et al., 2016), and COCO-Stuff (Caesar et al., 2018) are widely recognized semantic segmentation benchmarks. ADE20K is a large-scale scene parsing dataset comprising over 20,000 high-resolution images that span a diverse range of scenes and environments. Each image is densely annotated with 150 distinct semantic categories, encompassing objects, environments, and parts. Cityscapes focuses on the analysis and understanding of urban scenes. It includes 5,000 finely annotated high-resolution images captured from 50 cities across Germany. Each image provides pixel-level annotations across 19 semantic categories, covering typical urban elements such as roads, vehicles, and pedestrians. COCO-Stuff is a curated subset of the Microsoft COCO dataset, containing more than 10,000 images. Each image is annotated with 171 categories, including common objects, scenes, and background elements. Distinguished by its rich and fine-grained annotations, COCO-Stuff serves as a robust benchmark for evaluating the performance of semantic segmentation models.

### 4.1 IMPLEMENTATION DETAILS

To enhance the model's adaptability across diverse application scenarios, we introduce four SegRet variants of varying capacities: Tiny, Small, Base, and Large. The corresponding decoder parameter counts are 0.814M, 0.814M, 0.871M, and 2.607M, respectively. Each variant employs the same backbone as Vision RetNet (Fan et al., 2023), pre-trained on the ImageNet1K dataset (Deng et al., 2009). The decoder's channel dimensions $C$ for the four configurations are set to [256, 256, 256, 512], respectively. The SegRet model is implemented based on the MMSegmentation framework (MMSegmentation Contributors, 2020) and trained using four NVIDIA A40 GPUs. For data preprocessing,

images from the ADE20K and COCO-Stuff datasets are randomly flipped and cropped to a resolution of $512 \times 512$, while Cityscapes images are processed to $512 \times 1024$. Specifically, for the ADE20K dataset, the inputs of the Large variant are randomly cropped to $640 \times 640$. To ensure a fair comparison, we refrain from employing advanced training strategies such as OHEM (Shrivastava et al., 2016). Instead, we adopt the cross-entropy loss function in conjunction with the AdamW optimizer, using a learning rate of 0.0001 and a weight decay of 0.01. Batch sizes are configured as 16 for ADE20K and COCO-Stuff, and 8 for Cityscapes. Training is conducted for 160,000 iterations on ADE20K and Cityscapes, and 80,000 iterations on COCO-Stuff. Evaluation protocols are aligned with those used in Mask2Former(Cheng et al., 2022) to ensure consistency and comparability.

## 4.2 MAIN RESULTS

We quantitatively analyze SegRet's results on ADE20K, CityScape, and COCO-stuff, showcasing its remarkable performance in semantic segmentation tasks. It should be noted that all FLOPs are calculated at an input resolution of $512 \times 2048$.

**ADE20K**  Table 1 presents a comparative analysis of SegRet-Tiny against recent SOTA methods in terms of parameter count and mIoU. The results demonstrate that SegRet achieves superior performance among models with comparable parameter scales, attaining an mIoU of 49.39 with only 14.01M parameters. For example, SegRet matches the performance of Mask2Former (Swin-T) while using less than one-third of the parameters. Moreover, when compared to Vision Mamba (Vim-Ti), which has a similar parameter count, SegRet surpasses it by 9.7% in mIoU. Additional experimental results across various model scales are included in the Appendix for further reference.

Table 1: Comparison of the proposed SegRet-Tiny model on the ADE20K validation dataset. In comparison to SOTA methods, our Tiny variant exhibits notable advantages in both parameter efficiency and mIoU performance. "SS" and "MS" denote single-scale and multi-scale inference, respectively. The best-performing results are highlighted in bold for clarity.

| Method | Backbone | Decoder head | Image Size | #params | mIoU(SS) | mIoU(MS) | FLOPs |
|---|---|---|---|---|---|---|---|
| SenFormer (Bousselham et al., 2021) | R50 | | 512*512 | 55M | 44.4 | 45.2 | 179G |
| SenFormer (Bousselham et al., 2021) | R101 | | 512*512 | 79M | 46.9 | 47.9 | 199G |
| SegFormer (Xie et al., 2021) | MiT-B1 | | 512*512 | 13.7M | 42.21 | 43.1 | 15.9G |
| SegFormer (Xie et al., 2021) | MiT-B2 | | 512*512 | 27.5M | 46.5 | 47.5 | 62.4G |
| Vision Mamba (Zhu et al., 2024) | Vim-Ti | UperNet | 512*512 | 13M | - | 40.2 | - |
| Vision Mamba (Zhu et al., 2024) | Vim-S | UperNet | 512*512 | 46M | - | 44.9 | - |
| SeMask (Jain et al., 2023) | SeMask Swin-T | FPN | 512*512 | 35M | 42.06 | 43.36 | 40G |
| SeMask (Jain et al., 2023) | SeMask Swin-S | FPN | 512*512 | 56M | 45.92 | 47.63 | 63G |
| Swin (Liu et al., 2021) | Swin-T | UperNet | 512*512 | 60M | - | 46.1 | 236G |
| Swin (Liu et al., 2021) | Swin-S | UperNet | 512*512 | 81M | - | 49.3 | 259G |
| RMT (Fan et al., 2023) | RMT-T | FPN | 512*512 | 17M | - | 46.4 | 33.7G |
| RMT (Fan et al., 2023) | RMT-S | FPN | 512*512 | 30M | - | 49.4 | 180G |
| SenFormer (Bousselham et al., 2021) | Swin-T | | 512*512 | 59M | 46 | - | 179G |
| MaskFormer (Cheng et al., 2021) | Swin-T | | 512*512 | 42M | 46.7±0.7 | 48.8±0.6 | 55G |
| Vmamba (Liu et al., 2024b) | VMamba-T | UperNet | 512*512 | 55M | 47.3 | 48.3 | 939G |
| Mask2Fromer (Cheng et al., 2022) | Swin-T | | 512*512 | 47M | 47.7 | **49.6** | 74G |
| **SegRet-Tiny** | RMT-T | | 512*512 | 14.01M | **48.76** | 49.39 | 72.28G |

**Cityscapes**  As presented in Table 2, our SegRet-Tiny model demonstrates strong performance on the Cityscapes dataset. In comparison with methods such as EFCD-Small (R101) and SegFormer (MiT-B1), SegRet achieves notable improvements in both SS and MS inference, attaining 81.75% and 82.17%, respectively, while maintaining a comparable parameter count. Furthermore, when contrasted with larger models such as SeMask and Segmenter(ViT-L/16 with Seg-L-Mask/16), SegRet delivers superior mIoU(MS) scores, highlighting its robust generalization capability in MS settings. Additional performance metrics for other model scales are provided in the Appendix.·

**COCO-Stuff**  The proposed SegRet-Tiny model exhibits substantial performance advantages on the COCO-Stuff dataset. As shown in Table 3, SegRet-Tiny outperforms several SOTA methods, including MaskFormer, SenFormer, SeMask, and APPNet, achieving higher mIoU scores. Specifically, the model attains mIoU scores of 42.22 and 43.32 under SS and MS inference settings, respectively. These results underscore SegRet-Tiny's strong performance and parameter efficiency. Additional performance comparisons across different model scales are provided in the Appendix.

Table 2: Comparison of the proposed SegRet-Tiny model on Cityscapes dataset.

| Method | Backbone | Decoder head | Image Size | #params | mIoU(SS) | mIoU(MS) | FLOPs |
|---|---|---|---|---|---|---|---|
| SenFormer (Bousselham et al., 2021) | R50 | | 512*1024 | 55M | 78.8 | 80.1 | 179G |
| SenFormer (Bousselham et al., 2021) | R101 | | 512*1024 | 79M | 79.9 | 81.4 | 199G |
| ECFD-tiny (Zhang et al., 2024a) | R50 | | 512*1024 | 41M | 79.91 | 81.18 | 206G |
| ECFD-small (Zhang et al., 2024a) | R50 | | 512*1024 | 51M | 80.14 | 81.32 | 222G |
| ECFD-tiny (Zhang et al., 2024a) | R101 | | 512*1024 | 60M | 80.5 | 81.48 | 245G |
| ECFD-small (Zhang et al., 2024a) | R101 | | 512*1024 | 70M | 80.74 | 82 | 261G |
| SeMask (Jain et al., 2023) | SeMask Swin-T | FPN | 768*768 | 34M | 74.92 | 76.56 | 84G |
| SegFormer (Xie et al., 2021) | MiT-B1 | | 1024*1024 | 13.7M | 78.5 | 80 | 243.7G |
| Segmenter (Strudel et al., 2021) | DeiT-B/16 | Seg-B*/16 | 768*768 | - | - | 80.5 | - |
| Segmenter (Strudel et al., 2021) | DeiT-B/16 | Seg-B*-Mask/16 | 768*768 | - | - | 80.6 | - |
| Segmenter (Strudel et al., 2021) | ViT-L/16 | Seg-L/16 | 768*768 | - | - | 80.7 | - |
| Segmenter (Strudel et al., 2021) | ViT-L/16 | Seg-L-Mask/16 | 768*768 | - | 79.1 | 81.3 | - |
| **SegRet-Tiny** | RMT-T | | 512*1024 | 14.01M | **81.75** | **82.17** | 72.28G |

Table 3: Comparison of the proposed SegRet-Tiny model on COCO-Stuff dataset.

| Method | Backbone | Decoder head | Image Size | #params | mIoU(SS) | mIoU(MS) |
|---|---|---|---|---|---|---|
| MaskFormer (Cheng et al., 2021) | R50 | | 640*640 | - | 37.1±0.4 | 38.9±0.2 |
| SenFormer (Bousselham et al., 2021) | R50 | | 512*512 | 55M | 40 | 41.3 |
| SeMask (Jain et al., 2023) | SeMask Swin-T | FPN | 512*512 | 35M | 37.53 | 38.88 |
| APPNet (Zhu et al., 2023) | HRNet-W48 | APPNet+HRNet | 520*520 | 69.7M | 36.9 | - |
| APPNet (Zhu et al., 2023) | HRNet-W48 | APPNet+OCR | 520*520 | 72.3M | 40.3 | - |
| **SegRet-Tiny** | RMT-T | | 512*512 | 14.01M | **42.22** | **43.32** |

**Qualitative analysis** As illustrated in Figure 3, we performed a qualitative analysis on the ADE20K validation set, conducting a detailed comparison between our SegRet-Tiny model and the Mask-Former (Swin-T) model. The results reveal that SegRet-Tiny consistently outperforms MaskFormer, particularly in capturing fine details and reducing classification errors. This performance advantage can be largely attributed to the robust feature extraction capabilities of Vision RetNet, combined with the simplicity and effectiveness of our proposed lightweight decoder architecture.

## 4.3 ABLATION STUDIES

In this section, we present a series of ablation studies designed to evaluate the effectiveness of the proposed residual structure in the decoder, with particular emphasis on the role of zero-initialized residual layers. Furthermore, we investigate the influence of the decoder's channel size $C$ on model performance. All experiments were conducted using the ADE20K dataset.

**Zero-initialized residual layer** Table 4 presents an investigation into the effect of incorporating zero-initialized residual (ZIR) layers within the proposed decoder architecture. The experiments were conducted using the SegRet-Small variant, with all configurations kept consistent with the standard training setup. Results indicate that the inclusion of the ZIR layer yields a 0.79% improvement in mIoU, while introducing only a modest parameter increase of 0.25M These findings underscore the efficacy of the SegRet decoder design.

Table 4: Influence of zero-initialized residual layers on SegRet

| Method | ZIR Layer | #params | mIoU(SS) |
|---|---|---|---|
| SegRet-Small | | 26.27M | 49.9 |
| | ✓ | 26.52M | 50.69 |

**The impact of the decoder channel size** $C$ We examined the influence of the decoder channel size $C$ on the performance of the proposed model. As shown in Table 5, increasing $C$ generally leads to improved model performance, with the highest mIoU achieved when $C = 512$. However, further increasing $C$ to 768 results in both increased model complexity and a decline in mIoU, suggesting diminishing returns beyond a certain capacity. Based on empirical validations, we adopted $C = 256$

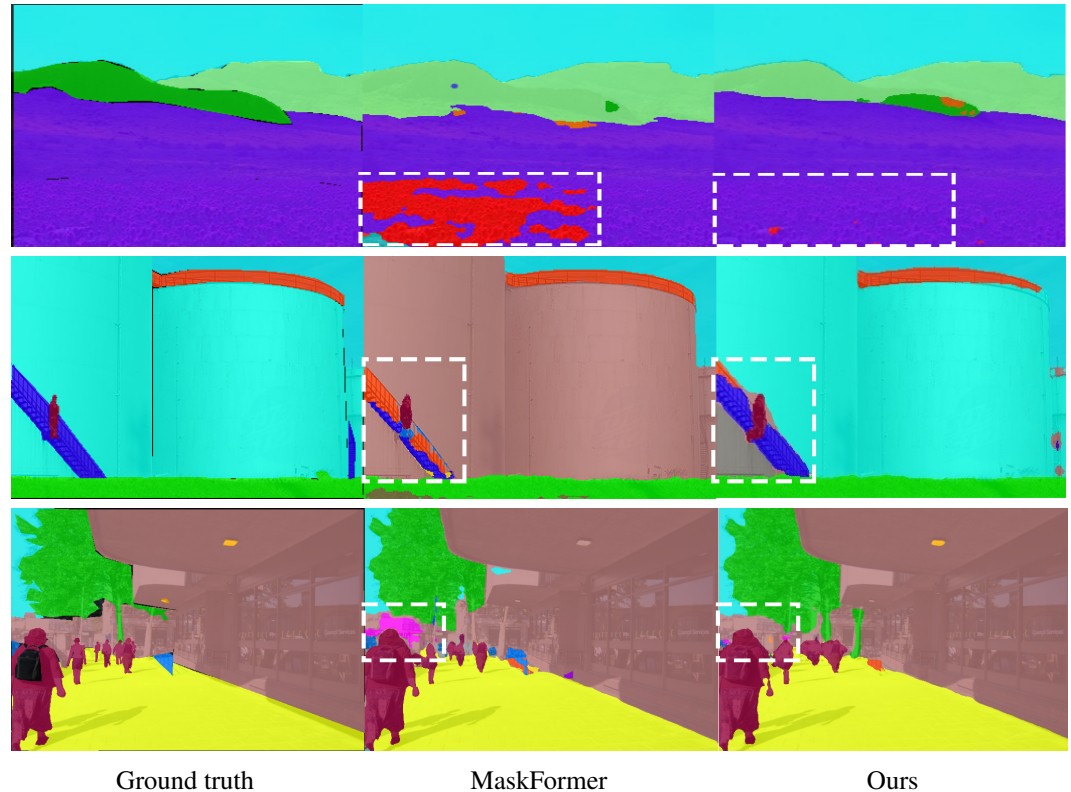

|                | Ground truth | MaskFormer | Ours |

Figure 3: **Qualitative analysis on the ADE20K.** The first column displays the ground truth values, while the outputs of MaskFormer and our proposed SegRet model are presented in the second and third columns, respectively.

for the Tiny, Small, and Base variants, and $C = 512$ for the Large model in our formal experimental settings.

Table 5: The Impact of Decoder Channel Size $C$ on SegRet

| Method | $C$ | #params | mIoU(SS) |
|---|---|---|---|
| | 256 | 94.26M | 50.9 |
| SegRet-Large | 512 | 95.81M | **52** |
| | 768 | 98.54M | 51.57 |

## 5 CONCLUSION

We introduce SegRet, a semantic segmentation model that integrates Vision RetNet as the encoder and employs a zero-initialized residual decoder. In this architecture, RetNet is leveraged for hierarchical feature extraction, while the decoder incorporates zero-initialized layers within its residual connections to enhance efficiency. Experimental results reveal that SegRet achieves remarkable performance across four variants and two benchmarks, maintaining or even improving segmentation accuracy despite a substantial reduction in parameter count. Nonetheless, a current limitation of SegRet lies in its underperformance on more specialized tasks such as medical image segmentation and remote sensing image analysis.

## REPRODUCIBILITY STATEMENT

We provide detailed descriptions of the SegRet architecture and training details in Section 3 and 4, with additional analyses in Appendix. An anonymous repository containing the full source code, dataset preprocessing scripts, and trained model configurations is available at `https://anonymous.4open.science/r/segret-D86C`, enabling reproduction of all reported results.

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

## A  ADDITIONAL RESULTS

In this section, we present a comprehensive analysis of the experimental results achieved by the SegRet-Small, Base, and Large variants on the ADE20K, Cityscapes, and COCO-Stuff datasets. Furthermore, we examine the effect of input resolution scaling on overall model performance.

### A.1  ADE20K

Table 6 summarizes the performance of models across different scales and architectural designs on the ADE20K validation set. Among smaller models, SenFormer and SegFormer deliver competitive results; however, our SegRet-Small variant surpasses both, achieving an mIoU of 50.7. In the mid-sized category, Swin and Mask2Former perform strongly, yet our SegRet-Base model attains a higher mIoU of 51.63. For large-scale models, MaskFormer and VMamba demonstrate commendable performance, with our SegRet-Large variant delivering results on par with these leading approaches.

Table 6: The results of various model sizes on the ADE20K dataset. * Indicates pretraining on ImageNet22K.

| Method | Backbone | Decoder head | Image Size | #params | mIoU(SS) | mIoU(MS) | FLOPs |
|---|---|---|---|---|---|---|---|
| SenFormer | Swin-S | | 512*512 | 81M | 49.2 | - | 202G |
| RMT | RMT-S | UperNet | 512*512 | 56M | - | 49.8 | 937G |
| RMT | RMT-B | FPN | 512*512 | 57M | - | 50.4 | 294G |
| SegFormer | MiT-B3 | | 512*512 | 47.3M | 49.4 | 50 | 79G |
| VMamba | VMamba-S | UperNet | 512*512 | 76M | 49.5 | 50.5 | 1037G |
| VMamba | VMamba-S | UperNet | 640*640 | 76M | **50.8** | 50.8 | 1620G |
| MaskFormer | Swin-S | | 512*512 | 63M | 49.8±0.4 | 51.0±0.4 | 79G |
| SegFormer | MiT-B4 | | 512*512 | 64.1M | 50.31 | 51.1 | 95.7G |
| **SegRet-Small** | RMT-S | | 512*512 | 26.52M | 50.7 | **51.29** | 117.1G |
| Swin | Swin-B* | UperNet | 640*640 | 121M | - | 51.6 | 471G |
| SegFormer | MiT-B5 | | 640*640 | 84.7M | 51 | 51.8 | 183.3G |
| RMT | RMT-B | UperNet | 512*512 | 83M | - | 52 | 1051G |
| Mask2Former | Swin-S | | 512*512 | 69M | 51.3 | **52.4** | 98G |
| **SegRet-Base** | RMT-B | | 512*512 | 53.05M | **51.63** | 52.13 | 229.66G |
| SeMask | SeMask Swin-B* | FPN | 512*512 | 96M | 49.35 | 50.98 | 107G |
| VMamba | VMamba-B | UperNet | 512*512 | 110M | 50 | 51.3 | 1167G |
| RMT | RMT-L | FPN | 512*512 | 98M | - | 51.4 | 482G |
| MaskFormer | Swin-B | | 640*640 | 102M | 51.1±0.2 | **52.3±0.4** | 195G |
| **SegRet-Large** | RMT-L | | 640*640 | 95.81M | **52** | 52.23 | 478.54G |

### A.2  CITYSCAPES

Table 7 provides a detailed comparison of model performance across various configurations on the Cityscapes dataset. In the small model category, our SegRet-Small model achieves mIoU scores of 82.59% (SS) and 83.26% (MS), outperforming other small-scale counterparts. For the base model group, SegRet-Base records 83.17% in SS and 83.8% in MS, delivering superior accuracy while maintaining moderate computational complexity. In the large model category, SegRet-Large attains an mIoU of 83.36% (SS), marginally outperforming comparable models. While it slightly trails Mask2Former (Swin-B*) in MS performance (84.5%), SegRet-Large maintains a parameter count of just 95.81M, offering a favorable balance between performance and computational efficiency.

### A.3  COCO-STUFF

As shown in Table 8, we further evaluated the performance of SegRet models with different sizes on the COCO-Stuff dataset. The results highlight the clear advantages of our proposed models, which employ RMT-S, RMT-B, and RMT-L as backbone networks. Specifically, our models achieve high average IoU scores of 44.32, 45.92, and 45.78 under single-scale (SS) inference, and 45.48, 46.06, and 46.63 under multi-scale (MS) inference, respectively. In contrast, competing models demonstrate comparatively lower performance across both evaluation settings. These results underscore the strong segmentation accuracy and generalization capabilities of our SegRet models on the COCO-Stuff dataset.

Table 7: The results of various model sizes on the Cityscapes dataset. * Indicates pretraining on ImageNet22K.

| Method | Backbone | Decoder head | Image Size | #params | mIoU(SS) | mIoU(MS) | FLOPs |
|--------|----------|--------------|-----------|---------|----------|----------|-------|
| Mask2Former | R50 | | 512*1024 | 44M | 79.4 | 82.2 | 293G |
| Mask2Former | R101 | | 512*1024 | 63M | 80.1 | 81.9 | 226G |
| SeMask | SeMask Swin-S | FPN | 768*768 | 56M | 77.13 | 79.14 | 134G |
| Mask2Former | Swin-T | | 512*1024 | 47M | 82.1 | 83 | 232G |
| SegFormer | MiT-B2 | | 1024*1024 | 27.5M | 81 | 82.2 | 717.1G |
| **SegRet-Small** | RMT-S | | 512*1024 | 26.52M | **82.93** | **83.52** | 117.1G |
| SeMask | SeMask Swin-B* | FPN | 768*768 | 96M | 77.7 | 79.73 | 217G |
| SegFormer | MiT-B3 | | 1024*1024 | 47.3M | 81.7 | 83.3 | 962.9G |
| Mask2Former | Swin-S | | 512*1024 | 69M | 82.6 | 83.6 | 313G |
| **SegRet-Base** | RMT-B | | 512*1024 | 53.05M | **83.28** | **83.87** | 229.66G |
| SeMask | SeMask Swin-L* | FPN | 768*768 | 211M | 78.53 | 80.39 | 455G |
| SegFormer | MiT-B5 | | 1024*1024 | 84.7M | 82.4 | 84 | 1460.4G |
| Mask2Former | Swin-B* | | 512*1024 | 107M | 83.3 | **84.5** | 466G |
| Mask2Former | Swin-L* | | 512*1024 | 215M | 83.3 | 84.3 | 868G |
| ECFD-tiny | Swin-Large | | 512*1024 | 209M | 82.67 | 83.41 | 473G |
| ECFD-small | Swin-Large | | 512*1024 | 218M | 83.1 | 83.61 | 488G |
| **SegRet-Large** | RMT-L | | 512*1024 | 95.81M | **83.36** | 83.91 | 478.54G |

Table 8: The results of various model sizes on the COCO-Stuff dataset. * Indicates pretraining on ImageNet22K.

| Method | Backbone | Decoder head | Image Size | #params | mIoU(SS) | mIoU(MS) |
|--------|----------|--------------|-----------|---------|----------|----------|
| MaskFormer | R101 | | 640*640 | - | 38.1±0.3 | 39.8±0.6 |
| MaskFormer | R101c | | 640*640 | - | 38.0±0.3 | 39.3±0.4 |
| SenFormer | R101 | | 512*512 | 79M | 41 | 42.1 |
| **SegRet-Small** | RMT-S | | 512*512 | 26.52M | 44.32 | 45.48 |
| SeMask | SeMask Swin-S | FPN | 512*512 | 56M | 40.72 | 42.27 |
| **SegRet-Base** | RMT-B | | 512*512 | 53.05M | **45.92** | 46.06 |
| SeMask | SeMask Swin-B* | FPN | 512*512 | 96M | 44.68 | 46.3 |
| **SegRet-Large** | RMT-L | | 512*512 | 95.81M | 45.78 | **46.63** |

## A.4 INPUT SCALING

We also conducted input scaling experiments using the Cityscapes dataset to examine the effect of varying image resolutions on model performance. As illustrated in Figure 4, our model consistently outperformed alternative methods across four different input sizes: $512 \times 1024$, $768 \times 768$, $640 \times 1280$, and $1024 \times 1024$. Notably, the model achieved its best performance with an input size of $1024 \times 1024$, reaching a peak mIoU of 82.02. Despite its competitive accuracy, our model maintains a compact design with only 14.01M parameters, highlighting its effectiveness in balancing performance and parameter efficiency.

## A.5 THE EFFECTIVENESS OF THE DECODER

As illustrated in Figure 5, we conducted experiments evaluating the performance of different decoder architectures when integrated with the RMT-T backbone. The results reveal that the RMT-T+UperNet achieves mIoU scores of 47.51 (SS) and 48.84 (MS), while the RMT-T+FPN combination yields 47.20 and 48.13, respectively. Notably, the integration of RMT-T with our proposed decoder achieves the highest performance, attaining mIoU scores of 48.76 (SS) and 49.39 (MS). These findings underscore the pivotal role of the decoder in enhancing the overall segmentation performance.

## A.6 LIMITATIONS

Although SegRet demonstrates robust performance in semantic segmentation of road scenes for autonomous driving, its generalizability to other domains like medical imaging and remote sensing

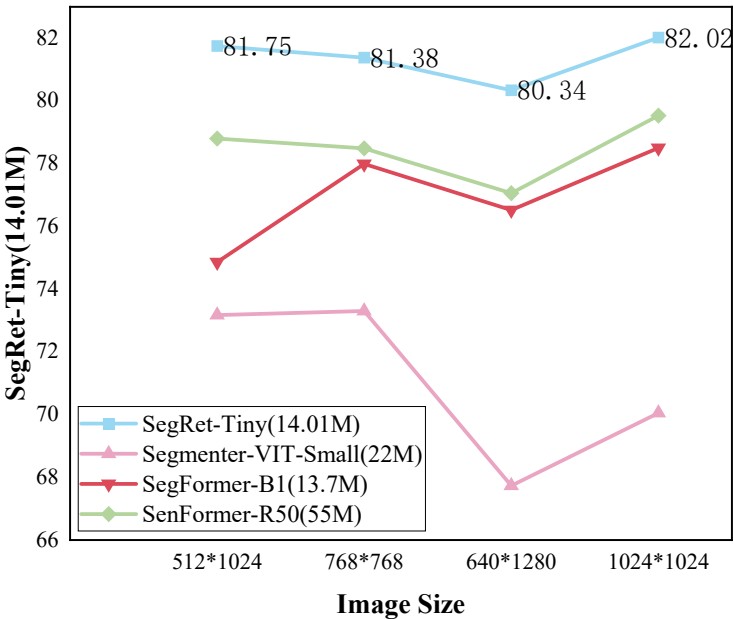

Figure 4: **A Comparative Study of Scaling Input Sizes on Cityscapes.** We examined the impact of four different input dimensions ($512 \times 1024$, $768 \times 768$, $640 \times 1280$, and $1024 \times 1024$) on the model performance.

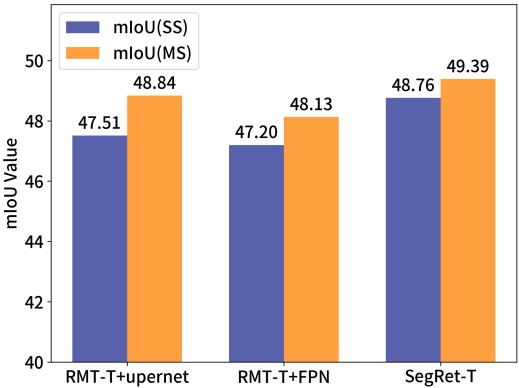

Figure 5: Performance Insights of Vision RetNet with Different Decoders.

remains limited. In medical image analysis, the model's ability to segment small structures is hindered by inadequate modeling of fine-grained features and cross-modal relationships. For remote sensing applications, substantial scale variations and complex scene compositions further challenge its adaptability. Future research directions may include developing domain-adaptive modules to enhance feature distribution alignment across different domains, as well as incorporating attention-guided upsampling mechanisms in the decoder to better capture small-object characteristics.

