# OpenReview forum: "SegRet: An Efficient Design for Semantic Segmentation with Retentive Network"
_ICLR.cc/2026/Conference — Submitted to ICLR 2026_

### Official Review · Reviewer_Qwqz · 2025-10-27

**Soundness:** 2
**Presentation:** 1
**Contribution:** 2
**Rating:** 2
**Confidence:** 4

**Summary:**

The paper introduces SegRet, a semantic segmentation model that pairs a RetNet backbone with a lightweight residual decoder using zero-initialization. The design targets the accuracy–efficiency trade-off common in autonomous driving and intelligent transportation. Specifically, the zero-initialized residual decoder keeps computation low while stabilizing information flow, and RetNet provides robust hierarchical feature extraction. The approach emphasizes parameter efficiency, claiming SOTA-level performance with markedly fewer parameters. Experiments on ADE20K, Cityscapes, and COCO-Stuff demonstrate strong accuracy under tight computational budgets, indicating SegRet’s suitability for real-time or resource-constrained deployments.

**Strengths:**

Balanced accuracy and efficiency: SegRet achieves strong segmentation performance on major benchmarks (ADE20K, Cityscapes, COCO-Stuff) while significantly reducing parameter count and computational cost, demonstrating excellent scalability for real-time or embedded applications.
Strong feature representation capability: By leveraging RetNet’s long-range dependency modeling, SegRet enhances multi-level feature extraction, resulting in richer contextual understanding and improved segmentation quality.

**Weaknesses:**

The contribution hinges on pairing RetNet with a zero-initialized lightweight residual decoder, but the paper doesn’t explain why this pairing is fundamentally new vs. existing efficient backbones + residual decoders, or how it differs from prior zero-init/ResNet-style stabilizers.

**Questions:**

1.The main text does not reference Figure 1. In addition, the meaning of multi-scale inference is unclear, and it should be explained why each method in the comparison has multiple parameter counts.
2.The Vision RetNet Backbone is an existing work, yet a disproportionate amount of space is devoted to describing it.
3.In Figure 2, the MASA diagram in the upper-left corner shows only one input, whereas the lower diagram depicts six inputs—this inconsistency should be clarified.
4.Equations (9–14) are densely stacked, which reduces readability and should be reformatted for clarity.
5.In Section 4.1, both Tiny and Small variants are reported to have identical parameter counts (0.814 M), which seems implausible and requires verification.
6.In Sections 4.2 and A.1–A.3, using dataset names as sub-section titles is inappropriate; more descriptive methodological headings are recommended.
7.In Table 3, the comparative methods are too few, especially recent ones from the past two years.
8.In Figure 3, the qualitative comparison on the ADE20K dataset includes only one baseline, and noticeable differences remain between the proposed results and the ground truth. Moreover, qualitative comparisons on the other two datasets are missing.
9.The description of the zero-initialized component is vague—does it consist of only a single convolution operation?
10.The overall contribution lacks substantial innovation beyond architectural integration.

---

### Official Review · Reviewer_2JaX · 2025-10-30

**Soundness:** 2
**Presentation:** 3
**Contribution:** 2
**Rating:** 2
**Confidence:** 4

**Summary:**

The paper proposes SegRet, a new semantic segmentation framework that integrates the recently developed Retentive Network (RetNet) — a state-space model designed to replace self-attention — with a lightweight, zero-initialized residual decoder, while significantly reducing parameter count, making it a compact, efficiency-oriented yet not fully state-of-the-art semantic segmentation framework.
This design enables SegRet to achieve competitive segmentation performance across ADE20K, Cityscapes, and COCO-Stuff using significantly fewer parameters and computational resources than traditional Transformer-based models like Mask2Former or Swin, making it particularly well-suited for real-time or resource-constrained vision applications such as autonomous driving and embedded systems.

**Strengths:**

Clean encoder–decoder story using Vision RetNet as a hierarchical backbone (Fig. 2, p. 4; Sec. 3.1) and a small decoder (Sec. 3.2, p. 5–6).

Competitive tiny/small regime: SegRet-Tiny (≈14 M params) is strong vs. other “tiny” setups across datasets (Tables 1–3, pp. 7–8).

Readable presentation and sensible training protocol (Sec. 4.1, p. 7), with an (anonymous) code link in the Reproducibility Statement (p. 10).

The principal strength of SegRet lies in its computational efficiency and architectural simplicity—it successfully integrates the Vision RetNet encoder, which captures long-range dependencies with linear complexity, with a lightweight zero-initialized residual decoder that minimizes parameters without degrading accuracy.

**Weaknesses:**

Vision RetNet is adopted largely as-is; the paper does not contribute new retention variants for vision, nor new theory atop RetNet. The retention machinery and bidirectional vision adaptation (BiRetention, horizontal/vertical decomposition) are recaps of prior work (Sec. 3.1; Eqs. 6–14).

The “zero-initialized residual decoder” amounts to a zero-init 1×1 residual branch after channel unification, followed by standard upsample-concat-conv (Eqs. 15–18). This design is extremely close to well-known lightweight decoders (FPN-style merges, 1×1 residual adapters), and the paper does not show conceptual novelty beyond the initialization trick.

Impact of the “novelty” is marginal. The only ablation directly tied to the claimed novelty (adding ZIR) shows +0.79 mIoU with +0.25M params on ADE20K for the Small variant (Table 4), which is a small incremental gain that does not justify SOTA claims by itself.

The core claim is “parameter/efficiency superiority,” yet there is no latency (ms), throughput (FPS), or peak memory on any GPU/CPU, nor profile at multiple resolutions (Sec. 4, pp. 7–9). FLOPs alone do not predict wall-clock; kernel efficiency (e.g., attention vs. state-space primitives) and cache behavior matter. Absent these, “efficient” remains asserted, not demonstrated.


Provide a theoretical or empirical rationale for the zero-init residual path: gradients at init, linearization analysis, effect on optimization dynamics; compare with ResNet-style identity and LayerScale variants.

**Questions:**

1. The “zero-initialized residual” decoder seems to differ from prior FPN/UPerNet-style decoders only by a zero-initialized 1×1 residual branch. Could you provide a formal motivation or derivation—for instance, how zero initialization affects gradient flow, optimization stability, or representational bias compared to standard residual or skip connections?

2. How is SegRet conceptually distinct from known efficient decoders such as SegFormer’s MLP-based fusion, LiteSeg, or MobileViT decoders? Please clarify what architectural element or training principle is genuinely new, rather than a simplified re-implementation.

3. Since The SegRet model is implemented based on the MMSegmentation framework could you highlight key difference between both of them.

4. You state the method underperforms for medical/remote sensing (p. 9; A.6, p. 15). Can you include one focused study (e.g., small-lesion segmentation) with failure analysis to clarify whether the limitation is encoder scale, decoder capacity, or upsampling choice?

---

### Official Review · Reviewer_uTUL · 2025-10-31

**Soundness:** 2
**Presentation:** 3
**Contribution:** 1
**Rating:** 2
**Confidence:** 5

**Summary:**

This paper proposed a semantic segmentation network that leverages retentive network. The network consists of a encoder backbone and a decoder. The encoder backbone contains 4 consecutive RMT blocks. The decoder utilizes a zero initialized convolution layer in parallel with a linear layer. Experiments on 3 benchmark datasets ADE20K, Cityscapes and COCO-Stuff and demonstrate the effectiveness of the proposed method.

**Strengths:**

The paper is well organized. The proposed methods achieves SOTA performance with low computational resources.

**Weaknesses:**

1. Lack of contribution. The proposed method's backbone is almost the same as the compared method RMT, which cannot be viewed as contribution. The improvement in the decoder is also minimum.

2. The compared methods contains mostly general vision backbones, and the result on many other vision tasks, such as object classification and detection, are available. But the proposed method is not comparing with them.

3. The comparison in Tab. 2, 3, 6, 7, 8 is not fair. The image size should keep the same, because the number of parameters, latency and losses are calculated depends on the image size.

4. More qualitative results should be provided, such as more segmentation results from compared methods, feature analysis, etc.

**Questions:**

See weakness.

---

### Official Review · Reviewer_eeLP · 2025-11-01

**Soundness:** 2
**Presentation:** 3
**Contribution:** 2
**Rating:** 6
**Confidence:** 3

**Summary:**

SegRet couples Vision RetNet as a hierarchical encoder with a lightweight zero-initialized residual decoder. It targets parameter efficiency while maintaining competitive mIoU on ADE20K, Cityscapes, and COCO-Stuff. Ablations show a modest but consistent gain from the zero-initialized residual (≈+0.8 mIoU) and comparisons demonstrate favorable accuracy/params trade-offs (e.g., SegRet-Tiny ~14M params reaching ~49.4 mIoU on ADE20K and ~42.2/43.3 mIoU on COCO-Stuff). Code link (anonymous) is provided.

**Strengths:**

1. This paper achieves a strong efficiency trade-off, particularly at smaller model sizes, with documented improvements over comparable baseline methods.

2. This paper provides transparent ablation studies on decoder design and input scaling, along with clear training details and code to ensure reproducibility.

3.  This paper honestly addresses limitations and suggests plausible next steps, such as domain adaptation and attention-guided upsampling.

**Weaknesses:**

1. The novelty lies mainly in the minimalist decoder design; the use of RetNet as the encoder and residual fusion represents an incremental improvement rather than a conceptual breakthrough.

2. While results are competitive, they do not clearly establish state-of-the-art performance under similar computational constraints in the most challenging benchmarks. Stronger comparisons using identical training configurations would be beneficial.

**Questions:**

1. The authors should provide FLOPs and latency comparisons with Mask2Former or SegFormer at the same input size to validate the efficiency claims.

2. Results on small-object segmentation (e.g., Cityscapes fine classes) are needed to evaluate if the lightweight decoder compromises fine detail.

---

### Meta-Review · Area_Chair_6AKb · 2026-01-06

**Summary:**

This paper proposes a lightweight semantic segmentation network with a hierarchical encoder and a zero-initialized residual decoder. All reviewers agree that the work lacks novelty and offers a limited contribution. In particular, the proposed backbone closely resembles existing architectures, with no clear conceptual advancement. The introduced components provide only incremental improvements, and the paper does not present sufficient theoretical or empirical justification for the design choices. Overall, the reviewers concur that the paper does not meet the novelty and significance requirements of this venue.

**Reviewer Concerns:**

There was no author response, so no concerns were addressed.

**Reviewer Scores:**

There was no author response, so the reviewers would not change their scores.

---

### Decision · Program_Chairs · 2026-01-26

Reject